# Control Effects of Short-Term Heatwaves on a Holocyclic Aphid

**DOI:** 10.3390/insects15020100

**Published:** 2024-02-01

**Authors:** Cirui Wu, Dailin Liu, Chengxu Gu, Zhenqi Tian, Xinxin Zhang, Jian Liu

**Affiliations:** College of Plant Protection, Northeast Agricultural University, Harbin 150030, China; wcr19991112@163.com (C.W.); ldl535914@163.com (D.L.); 18346255698@163.com (C.G.); tzq152519@163.com (Z.T.); xinxinz@neau.edu.cn (X.Z.)

**Keywords:** *Aphis glycines*, development, heatwave, morphogenesis

## Abstract

**Simple Summary:**

When holocyclic aphids, *Aphis glycines* Matsumura, were exposed to a diurnal of 35 °C lasting for seven days, the development, reproduction, and morphogenesis were affected significantly. These results are important for predicting the dynamics of *A. glycines* in this region. This also provides evidence that shows short-term heatwaves are useful for controlling the occurrence of *A. glycines*.

**Abstract:**

The soybean aphid, *A. glycines*, is an important soybean pest. Harbin, in the Heilongjiang Province, is an area with increasing temperatures in China that faces frequent short-term heatwaves. In this study, the development, reproduction, and morph differentiation of *A. glycines* have been studied when they were exposed to diurnal at 35 °C for seven days, beginning at different developmental stages. The nymph stage duration of *A. glycines* was longer, the adult lifespans and total lifespans were shorter, and their bodies were smaller when exposed to a diurnal of 35 °C beginning at the 1st to 4th stadium. The adult reproduction period was shorter, and the adult fecundity and intrinsic rate of increase were smaller than those of aphids reared at a diurnal of 25 °C. A higher and lower proportion of gynoparae and males were deposited as offspring on day 13 by adults when exposed to diurnal at 35 °C, beginning at the 1st to 4th stadium, respectively, than those of aphids reared at a constant of 20 °C. These results are important for predicting the dynamics of *A. glycines* in Harbin soybeans. This provides evidence that short-term heatwaves are probably useful for controlling *A. glycines*, by inhibiting development and male morphogenesis.

## 1. Introduction

The soybean aphid, *Aphis glycines* Matsumura, is native to eastern and southeastern Asia, and is commonly found in soybeans from China, Japan, the Korean Peninsula, Thailand, Malaysia, and Indonesia [1,2]. Soybean aphids mainly accumulate and feed on the tender stems or leaves of the soybean, where they suck the phloem sap. When soybeans are infested with *A. glycines*, they display symptoms such as leaf shrinkage, internode shortening, and plant dwarfing [3]. *Aphis glycines* can also transmit certain plant-viral species [4,5,6,7]. Serious damage to soybeans is caused by the soybean mosaic virus, and soybean losses of up to 50% have been reported [8]. Outbreaks of *A. glycines* occur sporadically in certain regions. In 2004, a heavy *A. glycines* infestation occurred in Suihua, Heilongjiang Province, China, causing yield losses of up to 30% and a total yield reduction of up to 112.5 million kilograms [9].

The life cycle of *A. glycines* is heteroecious and holocyclic. The winter hosts of *A. glycines* are buckthorn, *Rhamnus* spp., and the summer hosts are soybean, *Glycine max* (L.) Merrill, and the wild soybean species, *Glycine soja* Sieb and Zucc [10]. It overwinters as eggs under the buds of the buckthorn branches. When the temperatures rise to 10 °C in the spring, overwintering eggs hatch into wingless fundatrices, and winged virginoparae are produced, which migrate into the soybean. During the summer, *A. glycines* virginoparae can colonize soybeans and reproduce parthenogenetically throughout the season. As the temperature decreases and the photoperiod shortens in autumn, winged gynoparae are produced in the field and migrate to the buckthorns, where they produce oviparae. Winged males develop in the field and migrate to buckthorns, where they mate with oviparae, which lay overwintering eggs [2,11,12].

Global warming is an unprecedented phenomenon. Extreme weather events, such as heavy rainfall, drought, and heatwaves occur with an increasing frequency as global warming intensifies [13]. Heatwaves are weather events with temperatures above 35 °C, lasting for more than three days [14]. Extremely high temperatures are harmful to insects, causing sterility in various males [15], destroying the immune system [16], and reducing their resistance to pesticides [17]. Even if the temperature exceeds the higher developmental threshold of the insects for a short period of time, the cell water balance is destroyed [18], and the pH and ion concentration levels of the cells are considerably affected. As a result, some biological macromolecules lose their activity [19], which eventually leads to the impairment of bodily functions or death. High-latitude regions are generally more sensitive to global warming than low-latitude ones. Harbin in Heilongjiang, located at the mid- and high latitudes of the middle temperate zone, is subjected to considerable climatic changes, such as rising temperatures [20]. In this region, the local temperature is rising at a rate of 0.37 °C per decade [21]. In recent years, diurnal high temperatures above 35 °C, lasting for approximately 7 days have been occurring frequently during summer and autumn in this region [22]. In Harbin, the soybeans, *G. max*, are widely planted as food and oil crops, and the soybean aphid, *A. glycines*, is an important soybean pest. In the context of frequent heatwaves, it is unclear whether this pest needs to be controlled in this region.

As mentioned previously, there are multiple morphs in the life cycle of *A. glycines*. In the summer, *A. glycines* virginoparae colonize the soybean, and nymphs at different stages and adults coexist in the field. During heatwaves, both nymphs and adults face significant survival pressure. It is unclear whether the development and reproduction of *A. glycines* virginoparae at different developmental stages are considerably affected by heatwaves. In autumn, the gynoparae and males are deposited by *A. glycines* virginoparae. In the context of heatwaves during this season, it is unclear whether the morph differentiation proportion of gynoparae and male offspring deposited by the virginoparae is significantly different. If the development and reproduction of *A. glycines* virginoparae or gynoparae and males’ morphogenesis are affected considerably, when exposed to heatwaves it will probably be difficult for *A. glycines* to complete its life cycle. To confirm the hypothesis, we performed the experiments in this study. The results of this study are important for predicting the dynamics of *A. glycines* in soybeans in Harbin, Heilongjiang Province, northeast China, where the local environmental temperature is increasing, and short-term heatwaves occur more frequently. They also provide a scientific basis to explain how short-term heatwaves control the occurrence of the holocyclic aphid, *A. glycines*.

## 2. Materials and Methods

### 2.1. Aphids and Soybean Plants

The soybean aphids, *A. glycines*, were collected from a soybean field at the Northeast Agricultural University (126.72° E, 45.74° N) in Harbin. The colony of *A. glycines* was reared asexually on soybean seedlings (Heinong 51) in a growth chamber at 25 ± 1 °C, 70 ± 5% relative humidity (RH), and with a photoperiod of 16L: 8D h with artificial light of 12,000 lux. To maintain *A. glycines*, 20–30 aphids were transferred from mature, heavily infested plants to young, non-infested plants every 2 weeks.

Seeds of *G. max* (variety Heinong 51) were purchased from Wuchang Fangyuan Agriculture Co., Ltd. (Wuchang, China). Peat soil was placed into a circular plastic pot (diameter by height = 10 × 10 cm), and six to eight seeds were planted in each pot. The pots were placed in a growth chamber at 25 ± 1 °C, 70 ± 5% RH, with a photoperiod of 16L: 8D h. Soybean seedlings with a length of 15–20 cm (V2–V3 stage) [23] were used for the experiment.

### 2.2. Survival, Development, and Reproduction of A. glycines Virginoparae Exposed to Short-Term Heatwaves Beginning at Different Developmental Stages

In total, 70 wingless adult *A. glycines* virginoparae were selected from the aforementioned colonies. They were reared on soybeans at a normal temperature (diurnal 25 ± 1 °C and nocturnal 20 ± 1 °C), 70 ± 5% RH, and at a photoperiod of 16L: 8D h in a growth chamber for a 24 h reproductive period; thereafter, all adults were removed using a small brush. Newly deposited nymphs were retained and reared in a chamber. A total of five heatwave treatments were set as follows: Each nymph was treated as a unit, with 50 nymphs per treatment. When the nymphs had developed into the 1st to 4th stadium and adult stage, respectively, they were placed into a growth chamber at a diurnal of 35 ± 1 °C and a nocturnal of 20 ± 1 °C, 70 ± 5% RH, and a photoperiod of 16L: 8D h. After 7 days, they were transferred back into a normal-temperature growth chamber. In the control treatment, 50 newly deposited nymphs of *A. glycines* were reared in a growth chamber at a diurnal of 25 ± 1 °C and a nocturnal of 20 ± 1 °C, 70 ± 5% RH, and a photoperiod of 16L: 8D h, throughout the trial. In this trial, the aphids were all reared individually using the moisturizing cotton method. Briefly, a piece of square, moisturized cotton (facial cleansing cotton; 2 × 2 × 0.4 cm, length × width × height) was placed on the bottom of a 45 mL, 4 × 4.5 cm (diameter × height) glass beaker, and a piece of round filter paper of 4.2 cm (diameter) was placed on the surface of the moisturized cotton. The filter paper was cut to a slightly larger size for firm fixing within the beakers. The filter paper and moisturized cotton were then wetted with 2200 μL of water via dropping of the water onto the surface of the filter paper with a pipette. Detached leaves of the *G. max* were cut into 1.5 cm^2^ square pieces using a pair of scissors. Each nymph was placed on the reverse side of a square piece of soybean leaf that adhered to the surface of the filter paper. A small drop of water was placed on the surface of the filter paper, and a square piece of soybean leaf was placed on the surface. Owing to the addition of slight pressure, using a small brush, adherence of the leaf to the filter paper occurred due to the surface tension of water [23,24]. Individual nymphs were examined daily for their survival and exuviation. When they had developed into adults, the nymphs deposited by each female were counted and removed. Adult longevity was recorded daily until death. Water was added to the surface of the filter paper every 7–10 days (400 μL each time). Leaves were replaced every 5–7 days, or when they became yellowish [25].

In the above trial, dead adults were kept in 75% alcohol solution. The body length and width of the adults were measured using an optical microscope and a micrometer; the minimum measurement scale score was 0.01 mm. To estimate the body size of the adult, superficial area (SA) was used, which was calculated as SA = *π* × *a* × *b*/4 (*π* ≈ 3.14, *a* = body length, *b* = body width) [26].

### 2.3. Morph Differentiation of the Offspring Deposited by A. glycines Virginoparae Exposed to Short-Term Heatwaves Beginning at Different Developmental Stages

Overall, 120 adults of *A. glycines* virginoparae were selected from the aforementioned colony and divided into six groups of 20 adults each. The aphids were reared in a growth chamber at a constant 25 ± 1 °C, 70 ± 5% RH, and a 16L: 8D h photoperiod for a 24 h reproductive period; thereafter, all adults were removed. The newly deposited nymphs in each group were placed into a low-temperature and short-photoperiod growth chamber (constant 20 ± 1 °C, 70 ± 5% RH, and a 10L: 14D h photoperiod) to induce the production of gynoparae and males in the offspring. Groups 1 to 5 were subjected to heatwaves. In these groups, when the nymphs had developed into the 1st to 4th stadium and adult stage, they were transferred to a growth chamber with a high temperature (diurnal 35 ± 1 °C and nocturnal 20 ± 1 °C), 70 ± 5% RH, and a 10L: 14D h photoperiod. After 7 days, they were transferred back into the low-temperature and short-photoperiod growth chamber. The 6th group comprised the control group. The newly deposited nymphs of *A. glycines* were reared at a constant 20 ± 1 °C, 70 ± 5% RH, and a 10L: 14D h photoperiod throughout the trial. The nymphs and adults were reared individually on soybeans using the moisturizing cotton method [24,25,27]. When the *A. glycines* nymphs in the 1st to 6th groups had developed into adults, the nymphs deposited on days 1, 7, and 13 were retained. When the nymphs had developed into adults, they were identified as virginoparae, males, or gynoparae, based on their morphological characteristics [28], and the proportion of morphs was recorded. Overall, three replicates were performed for each treatment group.

### 2.4. Data Analysis

Raw data of the nymph stage duration, adult lifespan, and adult fecundity, along with their means and standard errors of *A. glycines* exposed to diurnal 35 °C lasting for 7 days, beginning at different developmental stages, were calculated using the TWOSEX–MSChart software [29,30]. The adult reproduction duration refers to the period from the first to last day when an aphid reproduces. The means and standard errors of the intrinsic rate of increase in *A. glycines* were calculated by the bootstrap technique [31] using the program TWOSEX–MSChart [30].

The age-stage-specific survival rate (*s*_xj_, *x* = age, *j* = stage), age-specific survival rate (*l*_x_), age-stage-specific fecundity (*f*_xj_), and age-specific fecundity (*m*_x_) were calculated as follows [29]:$$s_{\mathrm{xj}}=\frac{n_{\mathrm{xj}}}{n_{01}}$$
$$l_{x}=\sum_{j=1}^{k}s_{\mathrm{xj}}$$
$$m_{x}=\frac{\sum_{j=1}^{k}s_{\mathrm{xj}}f_{\mathrm{xj}}}{\sum_{j=1}^{k}s_{\mathrm{xj}}}$$
where, *n*_01_ stands for the number of the first instar nymphs, and *k* stands for the number of stages. The intrinsic rate of increase (*r*), finite rate of increase (*λ*), mean generation time (*T*), and net reproductive rate (*R*_0_) was calculated as follows:$$\sum_{x=0}^{\infty }e^{-r(x+1)}l_{x}m_{x}=1$$
$$r=\frac{\ln R_{0}}{T}$$
$$\lambda =e^{r}$$
$$R_{0}=\sum_{x=0}^{\infty }l_{x}m_{x}$$
$$T=\frac{\ln R_{0}}{r}$$

Differences in nymph stage duration, adult lifespan, total lifespan (nymph stage duration + adult lifespan), adult reproduction duration, adult fecundity, and the intrinsic rate of increase in *A. glycines* among different developmental stages exposed to diurnal 35 °C were analyzed using the paired bootstrap test. Because bootstrap analysis uses random resampling, a small number of replications will generate variable means and standard errors; thus, 200,000 bootstrap iterations were used to reduce the variability of the results [32,33,34].

Differences in adult body size of *A. glycines* among different developmental stages were analyzed by an analysis of variance (PROC GLM) and Tukey’s honest significant difference (HSD) tests using SAS 8.1. The proportions of wingless virginoparae, gynoparae, and male offspring deposited by *A. glycines* on days 1, 7, and 13 were arcsine square root transformed for normal distribution, which were exposed to a diurnal of 35 °C beginning at the 1st to 4th stadium and at the adult stage. Differences in the proportions of these morphs among the developmental stages were analyzed with GLM and Tukey’s HSD test using SAS 8.1 [35].

## 3. Results

### 3.1. Survival, Development, and Reproduction of A. glycines Virginoparae Exposed to Short-Term Heatwaves, Beginning at different Developmental Stages

When *A. glycines* were exposed to diurnal 35 °C lasting for 7 days, beginning at the adult stage, the survival rate of nymphs was similar to that of the control. The survival rate of nymphs at 3–5 days decreased when *A. glycines* were exposed to diurnal 35 °C beginning at the 1st to 4th stadium. On the 7th day, ninety percentage of *A. glycines* developed into adults exposed to a diurnal of 35 °C beginning at adult stage or reared at the control temperature. Whereas more than 32% of the nymphs were still at nymph stages at the 7th day, when they were exposed to a diurnal of 35 °C beginning at different stadiums. The longest survival time of the nymphs was 11 to 12 days, when *A. glycines* were exposed to a diurnal of 35 °C beginning at the 1st to 4th stadium. When the aphids were exposed to a diurnal of 35 °C, beginning at the adult stage or reared at the control temperature, the longest survival time of nymphs was 8 to 9 days (Figure 1A). When *A. glycines* were exposed to diurnal 35 °C for 7 days, beginning at the 1st to 4th stadium, the survival rate of adults was lower than that of the controls. When the *A. glycines* adults were exposed to a diurnal of 35 °C, the survival rate before the 31st day was lower than that of the control. After the 32nd day, the survival rate of adults was similar to that of the control (Figure 1B).

When *A. glycines* were exposed to diurnal at 35 °C, beginning at the 1st to 4th stadium, the nymph stage duration and total lifespan were longer and shorter than those of the control, respectively. When *A. glycines* were exposed to a diurnal of 35 °C beginning at the adult stage, the nymph stage duration and total lifespan were as long as those of the control. When *A. glycines* were exposed to diurnal 35 °C beginning at the 1st to 4th stadium and adult stage, the adult lifespan was shorter than those of the control (Table 1).

When *A. glycines* were exposed to a diurnal of 35 °C, beginning at the 1st to 4th stadium and the adult stage, the adult reproduction duration was shorter than that of the control, and adult fecundity and intrinsic rate of increase were smaller than those of the control (Table 2).

When *A. glycines* were exposed to a diurnal of 35 °C, beginning at the 1st to 4th stadium, the adults were smaller than those of the *A. glycines* exposed to diurnal at 35 °C beginning at the adult stage, which were smaller than those of the control (*df* = 5, 214; *F* = 24.32; *p* < 0.0001) (Figure 2).

### 3.2. Morph Differentiation of Offspring Deposited by A. glycines Virginoparae When Exposed to Short-Term Heatwaves Beginning at Different Developmental Stages

When *A. glycines* were exposed to a diurnal of 35 °C, beginning at different developmental stages, the proportions of virginoparae and gynoparae (on day 1: *df* = 5, 17; *F* = 0.36; *p* > 0.05; on day 7: *df* = 5, 17; *F* = 8.42; *p* > 0.05) in offspring deposited by adults on day 1 and 7 were as high as those of the control (Figure 3A,B; Appendix A). When *A. glycines* were exposed to a diurnal of 35 °C, beginning at different developmental stages, a higher proportion of gynoparae and a lower proportion of males (*df* = 5, 17; *F* = 0.71; *p* < 0.05) were deposited as offspring on day 13 by adults, compared to the control (Figure 3C; Appendix A).

## 4. Discussion

In Harbin, Heilongjiang, Northeast China, soybeans are widely cultivated as food and oil crops, and *A. glycines* is an important soybean pest. When *A. glycines* is subjected to heatwaves in this region, it faces significant survival pressure. If the development, reproduction, and morph differentiation of *A. glycines* are considerably affected, this indicates that heatwaves probably have a pest control effect. This study has investigated the effects of heatwaves and has clarified the mechanisms by which heatwaves control the occurrence and performance of *A. glycines*.

When *A. glycines* nymphs were exposed to a diurnal of 35 °C lasting for 7 days, the adult lifespan and total lifespan were shorter than those of the control (Table 1). When *A. glycines* nymphs and adults were exposed to a diurnal of 35 °C, the adult reproduction duration was shorter than that of the control, and adult fecundity and intrinsic rate of increase were smaller than those of the control (Table 2). When *A. glycines* were exposed to a diurnal of 35 °C, beginning at different developmental stages, a higher proportion of gynoparae and a lower proportion of males were deposited as offspring on day 13 by adults when compared to the control (Figure 3C; Appendix A). These results led us to infer that heatwaves that last for seven days can probably control *A. glycines* efficiently, by inhibiting the development and reproduction of aphids and reducing the proportion of male offspring.

Adult fecundity and the intrinsic rate of increase in *A. glycines* at a diurnal of 25 °C in this study were lower when compared to those reported previously [36]. In this study, *A. glycines* was reared on detached soybean leaves, whereas the aphids used in this study were fed on living soybean plants [36]. Based on a previous study, the development and reproduction of *A. glycines* were affected by the rearing method applied [24]. The differences in adult fecundity and the intrinsic rate of increase in *A. glycines* in the aforementioned studies could be partially attributed to the different rearing methods used. The adult fecundity of *A. glycines* in the present study was lower than that reported previously [37]. In a previous study, *A. glycines* were reared on living soybean plants, and the aphids were collected from Shenyang, Liaoning Province, China [37]. In this study, *A. glycines* were collected from Harbin, Heilongjiang Province, China, and the differences in the adult fecundity could be partially attributed to the different geographical populations of the aphids. In this study, the adult lifespan of *A. glycines* was longer than those of the aphids kept at a constant 25 °C, and the intrinsic rate of increase was smaller [38], probably because of the different nocturnal temperatures in these trials. The nymph stage duration, adult lifespan, adult fecundity, and intrinsic rate of increase in *A. glycines* in the control of this study were similar to those reported for aphids reared at a diurnal of 25 °C and nocturnal at 20 °C [25]. In this study, gynoparae and males of *A. glycines* were deposited at 20 ± 1 °C and a 10L:14D h photoperiod in the control, which is consistent with the finding that gynoparae and males could be induced successfully under the conditions of a low temperature and a short photoperiod [39,40].

The body size of *A. glycines* adults is considerably affected by temperature [27]. With increasing temperature, it gradually decreases [41]. Aphids consume more energy for metabolism under high-temperature conditions, and only a small amount of energy is used to increase the body size. Thus, the aphid body size decreases at higher temperatures [42]. When *A. glycines* at different developmental stages were exposed to a diurnal of 35 °C lasting for 7 days, the body sizes of adults were smaller than those of the control (Figure 2), which is consistent with the aforementioned report. When *A. glycines* adults were exposed to a diurnal of 35 °C, their reproduction period was as long as that of aphids exposed to diurnal at 35 °C at the 2nd to 4th stadium, and the adult fecundity was as pronounced as that of aphids exposed to a diurnal of 35 °C at the 3rd and 4th stadium (Table 2). These aforementioned results contrast with the finding that nymphs of the rose grain aphid, *Metopolophium dirhodum*, have a higher temperature tolerance than their adults [43]. The adult reproduction period of *A. glycines* exposed to a diurnal of 35 °C at the 1st stadium was longer than that of individuals at the 2nd and 3rd stadium. The adult fecundity and intrinsic rate of increase in *A. glycines* exposed to diurnal at 35 °C at the 1st stadium were greater than those of *A. glycines* at the 2nd and 3rd stadium. However, it remains unclear whether the adaptability of *A. glycines* to high temperatures in the 1st stadium is better than that in the 2nd and 3rd stadium.

There are multiple morphs in the life cycle of *A. glycines*, including fundatrices (spring morph), virginoparae (summer morph), gynoparae, males, and oviparae (autumnal morph) [2,44]. This study showed that heatwaves lasting for 7 days had an impact on the development and reproduction of *A. glycines* virginoparae (Table 1 and Table 2) and on the morphogenesis of *A. glycines* gynoparae and males (Figure 3C). Although studies on the summer and autumn morphs of *A. glycines* have been conducted, the adaptability of fundatrices to a diurnal of 35 °C is still not clear. In future experiments, the development and reproduction of *A. glycines* fundatrices exposed to diurnal 35 °C conditions should therefore be studied. However, the biology of overwintered eggs deposited by oviparae is poorly understood. In case the total deposited numbers and hatching rate of overwintered eggs decrease at an exposure to diurnal at 35 °C for 7 days, there is evidence of control effects of short-term heatwaves on *A. glycines*.

Aphid morphogenesis is related to the exogenous melatonin levels. Melatonin secretion increases when insects are exposed to extreme temperatures, explaining their adaptability to temperature stress [45]. When pea aphids, *Acyrthosiphon pisum*, were treated with high concentrations of exogenous melatonin, the proportion of sexual morphs increased, and the dominant morph was male [46]. In the present study, a lower proportion of male offspring was deposited by *A. glycines* on day 13 when they were exposed to a diurnal of 35 °C at different developmental stages (Figure 3C). In future studies, aphids exposed to a diurnal of 35 °C should be examined, and the melatonin concentration should be determined. The lower proportion of deposited *A. glycines* males can most likely be explained by melatonin regulation. Therefore, a field survey on the dynamics of *A. glycines* gynoparae and males should therefore be conducted during autumn in Heilongjiang. If more gynoparae and fewer males of *A. glycines* can be found in the field during seasons with heatwaves when compared to those without heatwaves, it is likely that heatwaves inhibit male differentiation of *A. glycines*.

The soybean aphid *A. glycines* is common to all soybean-producing areas in China [47]. However, there were significant differences in the temperature adaptability of *A. glycines* among the different geographical populations [40]. Subsequently, extensive sampling should be conducted in the different regions of China in order to study the adaptability of *A. glycines* from different geographical populations to heatwaves. In this study, the *A. glycines* collected from soybeans were used instead of a monoclonal population. But how much genetic variation was present in the aphid population used here is unclear. If these experiments were repeated using a different starting colony with different genetics, the experiment might probably lead to different results. In the summer of 2022, many locations in China were hit by heatwaves that lasted for more than seven days [48]. In addition, it is necessary to investigate the effects of prolonged heatwaves on the development, reproduction, and autumnal-morph differentiation of *A. glycines* from different geographical populations. In nature, the appearance of high temperatures is a process of gradual heating, followed by gradual cooling. Consequently, future studies on the effects of high-fluctuating diurnal temperatures, mimicking natural conditions, on the development, reproduction, and morphogenesis of *A. glycines* should be conducted. This study on the development and reproduction of *A. glycines* exposed to heatwaves was conducted based on a single biological replicate, and then these life characteristics were analyzed using the bootstrap method. If more biological replicates could be used in experiments, it is also a good alternative experimental design.

## Figures and Tables

**Figure 1 insects-15-00100-f001:**
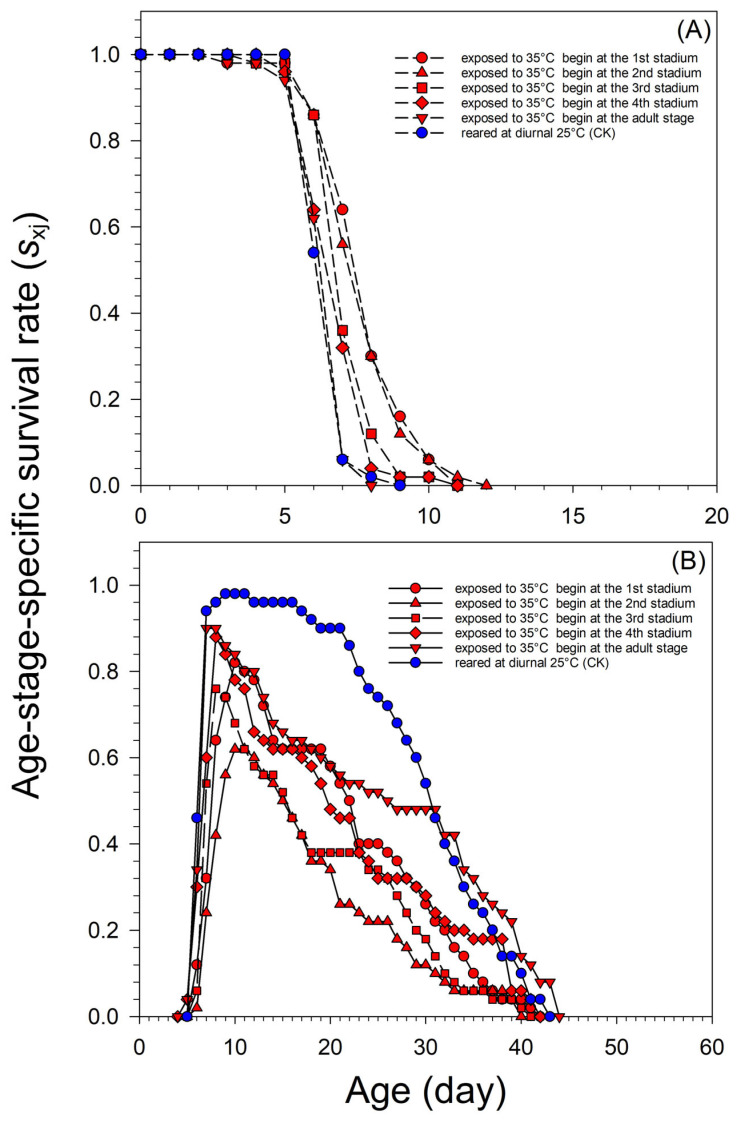
Age-stage-specific survival rate (*s_xj_*) of *A. glycines* virginoparae exposed to diurnal 35 °C for 7 days beginning at different development stages. To determine the difference in survival rate of nymphs, the maximum X-axis scales was set to 20. (**A**) *A. glycines* nymph (**B**) *A. glycines* adult.

**Figure 2 insects-15-00100-f002:**
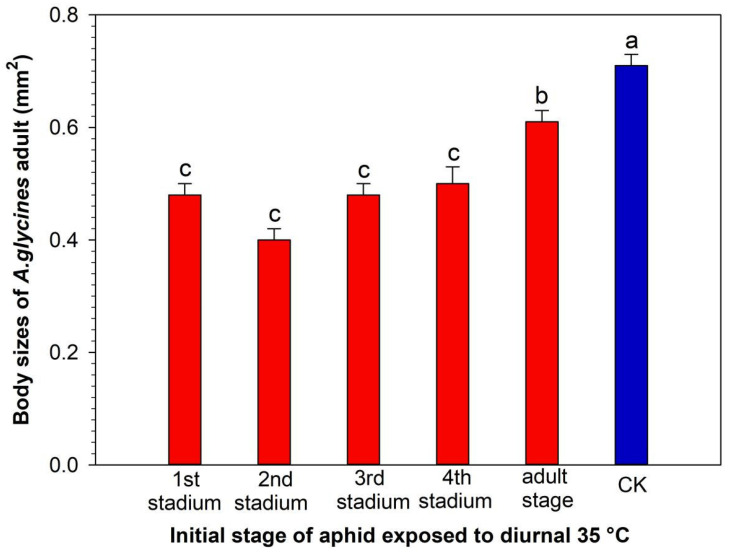
The body sizes of *A. glycines* virginoparae adults exposed to diurnal 35 °C for 7 days beginning at different developmental stages. CK—*A. glycines* were reared at diurnal 25 °C and nocturnal 20 °C throughout the trial. The columns marked with different letters are significantly different (Tukey’s HSD test, *p* < 0.05). Bars represent standard errors.

**Figure 3 insects-15-00100-f003:**
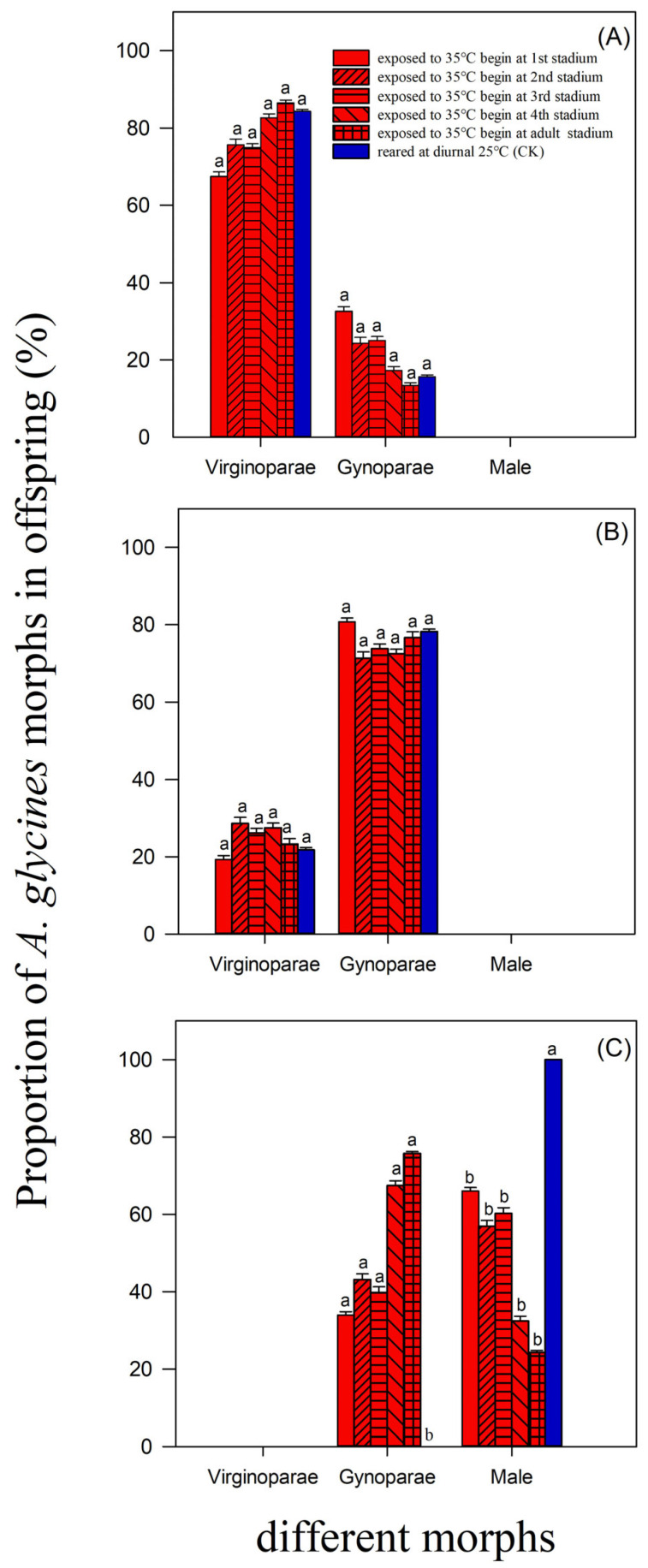
Proportion (Mean ± SE) of different morphs deposited by *A. glycines* on day 1, 7, and 13, which were exposed to diurnal 35 °C for seven days beginning at different developmental stages. (**A**) Morphs deposited on day 1, (**B**) Morphs deposited on day 7, (**C**) Morphs deposited on day 13.

**Table 1 insects-15-00100-t001:** Nymph stage duration, adult lifespan, and total lifespan (Mean ± SE) of *A. glycines* virginoparae exposed to diurnal 35 °C for 7 days beginning at different developmental stages.

Initial Stage of Aphids Exposed to Diurnal 35 °C	*n*	Nymph Stage Duration (Day)	Adult Lifespan (Day)	Total Lifespan (Day)
1st–stadium	45	8.00 ± 0.20 a	15.80 ± 1.39 bc	22.78 ± 1.45 bc
2nd–stadium	37	8.24 ± 0.21 a	12.43 ± 1.48 c	20.73 ± 1.50 cd
3rd–stadium	46	7.50 ± 0.14 b	12.30 ± 1.47 c	19.22 ± 1.47 cd
4th–stadium	46	6.98 ± 0.14 c	16.11 ± 1.58 bc	21.93 ± 1.60 bcd
Adult stage	49	6.65 ± 0.10 d	19.53 ± 1.82 b	24.76 ± 1.77 ab
Aphids reared at diurnal 25 °C (CK)	50	6.62 ± 0.09 d	23.68 ± 1.10 a	29.40 ± 1.10 a

The initial number of aphids used in each treatment was 50. The sample size (*n*) is the number of aphids that developed into adults, which were exposed to diurnal 35 °C for 7 days beginning at different developmental stages. Means within a column followed by the same letter do not differ significantly (paired bootstrap test, *p* < 0.05).

**Table 2 insects-15-00100-t002:** Adult reproduction duration, adult fecundity, and intrinsic rate of increase (Mean ± SE) in *A. glycines* virginoparae exposed to diurnal 35 °C for 7 days beginning at different developmental stages.

Initial Stage of Aphids Exposed to Diurnal 35 °C	*n*	Adult Reproduction Duration (Day)	Adult Fecundity (Offspring per Female)	Intrinsic Rate of Increase (Day^−1^)
1st–stadium	45	11.25 ± 1.03 b	19.91 ± 2.29 b	0.2055 ± 0.0093 d
2nd–stadium	37	5.81 ± 0.79 d	5.38 ± 1.10 d	0.1133 ± 0.0151 f
3rd–stadium	46	6.63 ± 0.85 d	9.98 ± 1.74 c	0.1782 ± 0.0102 e
4th–stadium	46	7.71 ± 0.84 cd	15.33 ± 1.96 b	0.2339 ± 0.0090 c
Adult stage	49	9.24 ± 0.70 bc	17.18 ± 1.26 b	0.2723 ± 0.0069 b
Aphids reared at diurnal 25 °C (CK)	50	15.62 ± 0.59 a	50.86 ± 1.94 a	0.3408 ± 0.0056 a

The initial number of aphids used in each treatment was 50. The sample size (*n*) is the number of aphids that developed into adults, which were exposed to diurnal 35 °C for 7 days beginning at different developmental stages. Means within a column followed by the same letter do not differ significantly (paired bootstrap test, *p* < 0.05).

## Data Availability

All data are presented within the article and Appendix A.

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
