# Peer review of "Control Effects of Short-Term Heatwaves on a Holocyclic Aphid"

_insects, 2024, doi:10.3390/insects15020100_

Round 1
Reviewer 1 Report
Comments and Suggestions for Authors
The Manuscript entitled, “Control Effects of Short ‒ Term Heatwaves on a Holocyclic Aphid” addressed a good question. Although it needs to improve with more scientific way especially language making it hard to understand which can be corrected in some place to just remove extra methodological information.
Most comments are included in the MS file. Some general comments are following:
Introduction is lacking hypothesis/objective
The figures quality needs to improve. Hard to follow in current form.
The results should be written more concisely.
Discussion part contains lot of irrelevant discussion or at least not suitable for discussion section instead can be used in other sections of MS.
Try not to repeat sentences among the paragraphs in discussion section. Discussion should have more implications of the results and can include regions where these results can be implemented.

Comments on the Quality of English LanguageThe writing is too complicated and not scientific most of time to understand so need thorough revision for language.
Reviewer 2 Report
Comments and Suggestions for Authors
This work examines interesting questions about the effect of elevated temperatures on the development of soybean aphids. Although the overall work is interesting and worth publishing, there are a number of areas that should be improved before publication.
- The experimental design needs to be clarified. This could be achieved by a diagram that shows how the different temperatures were applied, and when sampling and assessment occurred.
- How much genetic variation was present in the aphid populations used here? There is no mention of how many aphids comprised the starting population, and the use of 20 to 30 individuals for each colony transfer from a population reproducing parthenogenetically could lead to a very inbred population. If these experiments were repeated with a different starting colony with different genetics might lead to quite different results. This possibility is mentioned in the Discussion but mostly in the context of regional variation with A. glycines.
- What is the time for transitions between temperatures in these experiments?
- Lines 124-125 states “each nymph was treated as a unit, with 50 nymphs per treatment.”: I do not understand what this means. Does this refer to each nymph stage or something else?
- In the Discussion, reference is made to the use of separated soybean leaves, but this is not made explicit in the Methods. It would be helpful to clarify exactly how each exposure group was maintained during the experiment and a reference to the raising technique might be helpful.
- Data analysis needs considerable clarification. From what is stated here, it appears that a single replicate of each condition was used to collect data, and that means and standard errors were calculated using bootstrap techniques. If each exposure at each stadium was done just once, then these data should be treated very cautiously. Bootstrap approaches can certainly tell us something about the distribution of data within a replicate, but biological replicates are essential to establish how much biological variation might be present and this can’t be assessed using a bootstrap analysis of a single experiment.
- Figures 1 and 2 are difficult to assess because of their low resolution. These need to be higher resolution so the symbols and text are legible.
- In Figure 1, the Y-axis refers to “Age-stage-specific survival rate (sxj) but this is apparently not defined anywhere in the text.
- The data in Tables 1–5 form the basis of the findings: these data might be more usefully presented in graphical form which would make it easier to assess differences in treatments, with tabular data presented in Supplementary Material.
- Before accepting the overall findings of this work, the question of how many replicates are involved in generating the data needs to be answered. That would also help with understanding whether the findings hear which suggest that A. glycine are significantly affected by week-long diurnal periods of 35 °C are generally meaningful or specific to a particular batch of aphids. This is an important point since the authors note that considerable variation in heat tolerance as been reported for A. glycine different regions in China and world-wide.
Overall the quality of the writing is good and in most places, it seems clear what the authors intend. However, there are a number of places which could be improved and I think someone with a good understanding of idiomatic English could significantly improve the clarity of the manuscript.
- The Abstract needs to be clarified to make it clearer what was done. At present it is hard to identify clearly what was done and what was found.
- The use of “1st to 4th stadium” (Line 18 and elsewhere) does not make it clear that 1st to 4th stadium aphids were used in separate experiments.
- The use of hyphens (-) and dashes (–) is inconsistent and problematic in some cases. Phrases like “High-latitude” (Line 63 and elsewhere) should be hyphenated with gaps. Times should be expressed as “24 h” (Line 122) without a hyphen or dash.
- Line 269 (and elsewhere): “diurnal 35°C last for 7 days”: should this be “lasting for 7 days”?
- It is usual to specify genus and species the first time a binomial is used and to contract it thereafter. This should be tidied up in the manuscript.
Round 2
Reviewer 2 Report
Comments and Suggestions for Authors
Comments
The revised manuscript addresses most of the matters raised in my first report. These include a significant improvement to the writing and improved data presentation. There are still a few things that need attention.
- Survival, development, and reproduction … lines 130–166
- It is still unclear from what is written here exactly how many experimental replicates this work represents. As it stands, it appears to be a single experimental manipulation starting with 70 wingless adult virginoparae generating nymphs that were then assigned to one of five heatwave treatments along with a control treatment. If this the case, it needs to be explicitly stated as these results will need to be interpreted with care as they represent a single biological replicate.
- Morph differentiation of offspring … lines 167–187
- At the end of this section it is noted that these data represent three replicates (line 186): does this reflect three technical replicates of a single experiment, or three biological replicates?
- Age-stage-specific survival rate
- The information for deriving this is now provided and it is clearer how these values are calculated. However, the terms ‘R0’ and ’T’ are not defined (lines 206 and 209).
- Reference numbers
- During revision of the manuscript, some reference numbers have changed and there are a number of places where this has not been correctly updated. For example, lines 56, 67 and and 157.
- Figures 1 and 3
- Although these figures are more legible than previously, the version in the pdf that I downloaded remains low resolution and pixelated, and they do not scale with enlargement. In addition, much of the data plots close together and the large symbols that are used make it difficult to identify details of the treatments. Smaller symbols would help as would widening the graphs if there is space in the final manuscript. Either way, it would also be helpful to provide scaleable versions of these two figures in the supplementary data for those wishing to examine them closely.
- A minor point, but it seems that in Fig. 1, the x-axes of only panel A has been set to 0–20 (lines 268-269).
The discussion of proposed work in the Discussion is helpful and adds to the value of the manuscript.
